# Assessing Critical Flicker Fusion Frequency: Which Confounders? A Narrative Review

**DOI:** 10.3390/medicina59040800

**Published:** 2023-04-20

**Authors:** Thomas Muth, Jochen D. Schipke, Anne-Kathrin Brebeck, Sven Dreyer

**Affiliations:** 1Institute of Occupational, Social, Environmental Medicine, Faculty of Medicine, Heinrich-Heine-University Düsseldorf, 40225 Düsseldorf, Germany; 2Research Group Experimental Surgery, University Hospital Düsseldorf, 40225 Düsseldorf, Germany; 3Artemis Augenklinik, 60314 Frankfurt, Germany; 4Hyperbaric Oxygen Therapy, University Hospital Düsseldorf, 40225 Düsseldorf, Germany

**Keywords:** critical flicker fusion frequency, visual system, neurological diseases, alertness, cognition, diving, hyperbaric medicine

## Abstract

The critical flicker fusion frequency (cFFF) refers to the frequency at which a regularly recurring change of light stimuli is perceived as steady. The cFFF threshold is often assessed in clinics to evaluate the temporal characteristics of the visual system, making it a common test for eye diseases. Additionally, it serves as a helpful diagnostic tool for various neurological and internal diseases. In the field of diving/hyperbaric medicine, cFFF has been utilized to determine alertness and cognitive functions. Changes in the cFFF threshold have been linked to the influence of increased respiratory gas partial pressures, although there exist inconsistent results regarding this effect. Moreover, the use of flicker devices has produced mixed outcomes in previous studies. This narrative review aims to explore confounding factors that may affect the accuracy of cFFF threshold measurements, particularly in open-field studies. We identify five broad categories of such factors, including (1) participant characteristics, (2) optical factors, (3) smoking/drug use, (4) environmental aspects, and (5) breathing gases and partial pressures. We also discuss the application of cFFF measurements in the field of diving and hyperbaric medicine. In addition, we provide recommendations for interpreting changes in the cFFF threshold and how they are reported in research studies.

## 1. Introduction

The eyes are the primary organs responsible for receiving photons and initiating the visual pathway. This pathway encompasses a series of anatomical structures that convert light energy into electrical action potentials that can be interpreted by the brain. It starts at the retina and travels through various neural circuits before terminating at the primary visual cortex, which is located in the central nervous system (Figure 1).

When a light flickers above a certain frequency, the sensation of the flickering disappears, and the light appears to be continuous. The critical flicker fusion frequency (cFFF) is the threshold above which the receptor potential can no longer differentiate between changes. As a result, the potential waves combine to form a constant level of depolarization, which the central nervous system interprets as continuous light. Higher thresholds of cFFF have been linked to greater perceptual accuracy [1] as well as cortical arousal and visual processing [2].

Reports of cFFF in humans have varied depending on the respective participants and the equipment used, ranging from 22 to 90 Hz [3], 10 to 60 Hz [4], or 50 to 90 Hz [2]. However, normal adult cFFF is typically around 35 to 40 Hz [5], which agrees with the temporal resolution of the human photoreceptors in the central retina, limited to cFFFs below 50 Hz [6].

The observed large variations in cFFF thresholds may be attributed to the theory of signal detection [7], which allows us to differentiate between the participant’s ability to distinguish between events and motivational effects or response biases [8]. Although the motivation of patients in previous tests may have contributed to the large variation, this is difficult to assess. However, it can be assumed that participants in well-controlled open-field studies will be motivated and, as a result, cFFF thresholds can be more reliably measured.

Since 1956, cFFF has been used to investigate the electrical excitability of the retina [9]. More recently, it has been described as a tool to assess central nervous system fatigue and cortical arousal [10], as well as a metric indicative of fatigue [11], mental workload [12], and a measure for the assessment of alertness and cognitive performance [13,14]. It has even been used as a selection criterion for determining the physical fitness of professional divers [15]. Due to its simplicity and usefulness in measuring temporal resolution [16], cFFF tests are widely used for both clinical and research purposes [17].

For clinical purposes, cFFF is commonly used as a screening test for eye diseases [18] and is a well-established method for evaluating optic neuropathies such as optic neuritis and ischemic neuropathy [19]. For instance, healthy eyes have cFFF values of 31 Hz, while eyes with non-arteritic anterior ischemic optic neuropathy have a CFF value of 24 Hz. Eyes with demyelinating optic neuritis showed values significantly lower than 24 Hz, and thus, 24 Hz is a useful cutoff value when trying to distinguish between these two conditions [19].

Additionally, cFFF is used to assess digital eye strain [20], also known as computer vision syndrome [21], which occurs due to overuse of display screens and affects both the retina and central nervous system [22].

Furthermore, cFFF has been considered a useful tool for diagnosing age-related macular degeneration (AMD) [23] and glaucoma [24], but changes in cFFF may also be due to refraction errors and media opacity [25]. Although cFFF was found to be decreased in non-exudative and exudative AMD, it was not able to distinguish between eyes with either type of AMD versus healthy eyes with equal visual acuity. Therefore, cFFF testing was not deemed suitable as a diagnostic test of AMD by later researchers [26].

Nevertheless, cFFF can serve as a useful diagnostic test for numerous neurological and internal diseases [27]. For instance, it has been reported to aid in the early detection of Alzheimer’s disease [28] and hepatic encephalopathy [6,29,30].

However, caution is advised when using this method for patients diagnosed with minimal hepatic encephalopathy, as cFFF may give false positive results [31]. Despite the positive diagnostic potential of cFFF, the use of this method in clinical settings is not without its challenges.

For research purposes, cFFF has been extensively used to study the physiology of vision [32]. Therefore, it is logical to employ cFFF tests to estimate cognitive impairment or clinically relevant cerebral impairment [14]. For example, in the context of alcohol use [33,34]. one study reported a lowered cFFF threshold after moderate alcohol consumption [35]. Similarly, cFFF was statistically significantly lower in alcoholics than in healthy subjects in another study [36]. However, the cFFF threshold did not decrease further in alcoholics given 250 mL of vodka, suggesting that cFFF could be an objective measure of alcohol tolerance [37].

Sports divers usually do not consume alcohol before diving but may drink late in the evening on dive holidays or safaris. In cases where a hangover is present before the next day’s early dive, decreases in cFFF can be expected and may mask any effects of nitrogen narcosis.

Alternatively, cFFF has been used to assess cortical arousal above sea level in many studies. For example, during zero-gravity flights [38], parachute jumping [39], and in dry chambers [40,41,42] Below sea level, cFFF has been used in studies with apnoeic divers [43], scuba diving [44,45], and diving with a closed circuit rebreather (CCR) [14,46]. In particular, measuring cFFF during operational tasks below sea level has proven to be a simple, rapid, reliable, and reproducible tool nction have been attributed to the narcotic effect of increased nitrogen partial pressure (pN_2_) [40,44,47,48]. However, cFFF increased in a study that used trimix (5.5 bar) [14], and surprisingly decreased in another study (6 bar), that measured response speeds towards two reaction time tasks [49].

CFFF tests were also employed when investigating the effect of increased oxygen partial pressures (pO_2_). While normobaric O_2_ had no effect on cFFF, it decreased at a pO_2_ of 1.4 bar and increased at a pO_2_ of 2.8 bar. Changes in cFFF due to changes in pO_2_, therefore, appear to be dose-dependent [50]. Another study concluded that nitrogen (N_2_) and O_2,_ both alone and in combination, can induce neuronal excitability or depression in a dose-dependent manner [45].

Therefore, there are varying results regarding the influence of increased respiratory gas partial pressures. Additionally, a recent study [42] found that the cFFF was not suitable to record narcotic effects of respiratory gases in 6-bar dry-chamber conditions. However, this conclusion was questioned [51] due to the hyperbaric conditions being a mixture of N_2_ and O_2_ effects, further complicated by the presence of confounding factors.

This narrative review provides a comprehensive analysis of the critical flicker fusion frequency (cFFF) and the various factors that can confound its measurement. In doing so, it revisits the seminal work of Erlick and Landis (1952) [52] and expands on their observation that more than ten factors can influence the cFFF threshold. This study also seeks to address the specific confounders that are relevant to diving and hyperbaric medicine, where accurate measurement of cFFF is crucial. By examining these confounders, this review aims to enhance our understanding of the cFFF and its clinical applications in these fields. Ultimately, this analysis provides valuable insights for researchers and clinicians alike and can help improve the accuracy and reliability of cFFF measurements.

## 2. Confounding Factors

According to the literature, the “true” cFFF depends on a variety of parameters [52,53]. In this review, we use the term confounders and focus on several factors, including participants, smoking/drugs, optical aspects, environment and breathing gases.

### 2.1. Participants

Individuum: Our pilot study aimed to assess the intraindividual variations in cFFF in a single male adult over a period of 38 days. The measurements were always taken in the morning under consistent lighting conditions, with direct sunlight [54]. The result showed a cFFF threshold of 44.0 ± 1.7 Hz (mean ± SD). Such variability of approximately 4% of the cFFF threshold indicates that any interventions in future studies should induce changes greater than 4%.

Gender has not been found to effect mean cFFF values significantly between male and female adults [55], a result that was confirmed by our pilot study [54]. However, a moderate difference of 6% was found in one study among a group of 1000 adults, with males having slightly higher cFFF values than females [22].

In contrast, lower mean cFFFs were found in boys than girls, and children with problems (i.e., antisocial behaviors and attitudes, extraverted, maladjusted, or otherwise exhibiting traits of psychiatric behavioral problems) obtained significantly lower mean cFFFs than children who did not exhibit behavioral problems [56]. Therefore, cFFF has been suggested as an index of certain mental abilities in children [57], and we suggest that it could become an additional instrument to assess fitness to dive in the growing group of diving children due to its simplicity, rapidity, and non-invasive nature.

Age: Visual acuity tends to remain relatively constant from ages 40 to 50. However, it steadily declines as a person ages above 60 and on to the age of 80 [58]. Additionally, cFFF has been found to decrease with age [22,59], partly due to a reduction in retinal illumination [60]. Interestingly, cFFF has been shown to be a unique predictor of executive dysfunction, accounting for unique variance in performance above and beyond age and global cognitive status, across both younger and older age groups [61].

Personality: Although there are various instruments to investigate a person’s properties, the cFFF test has also been used to study the relationship between personality and frequency thresholds. The test has revealed that the thresholds in adults are personality-dependent [22]. For example, cFFFs for extraverts were shown to be lower than introverts [62]. Additionally, cFFF differences in learning behaviour depended on the subject’s level of sociability [63], while cFFF and academic performance had only a moderate correlation [64]. Interestingly, cFFF was found to be higher in adults who played computer or video games compared to those who did not. Furthermore, adults who played “instructive” games had a higher cFFF than those playing “quest type” games, although the types of games were not further defined beyond this categorization [22].

### 2.2. Smoking/Drugs

Smoking: Previous studies have found that smoking cigarettes leads to a raised cFFF threshold [35]. Therefore, divers who smoke a cigarette before a dive may exhibit an elevated cFFF that will spontaneously decrease during the dive. Such a decrease at any given depth may be erroneously interpreted as a nitrogen-induced effect.

Drugs: In a review article, amphetamines were described as improving cognition when used at therapeutic doses in healthy adults [65], and the cFFF test can therefore be regarded as a measure of cortical arousal. For instance, a single low dose of amphetamine led to an increase in the cFFF [66]. Similarly, cFFF thresholds after other CNS stimulants (pemoline, methylphenidate, hydergine, and chlorpromazine) were increased in parallel with self-ratings of alertness. Interestingly, clobazam, an anxiolytic, increased the cFFF in subjects who rated themselves as having high anxiety, but decreased cFFF thresholds in subjects who did not [17]. However, a single oral dose of diazepam decreased cFFF [66], while zolpidem, given before sleep, had no effect on the subsequent daytime cFFF [67]. Although the aforementioned drugs are unlikely to be consumed in the diving community, cannabis may begin playing a more significant role due to being legalized. A study showed that smoking 1 g of marijuana (1.5% THC) clearly increased the cFFF threshold [32]. Therefore, if a subject’s cFFF is assessed shortly after drug consumption, the cFFF will likely be erroneously increased. As a result, the cFFF thresholds of any delayed intervention will be overshadowed by the decreasing drug effect, making drug consumption likely to confound the cFFF thresholds.

### 2.3. Optical Factors

Vision: The fovea centralis is a small pit located in the central region of the eye that contains densely packed cones. This area is responsible for providing clear central vision, which is crucial for tasks that require visual precision, such as reading and driving. As one moves further away from the fovea centralis, visual acuity decreases [68].

The relationship between the cFFF threshold and retinal location is primarily determined by the density distribution of receptor cells (cones and rods) on the retina [69]. When a light source flickers, the amount of flicker is greater when its image falls on the fovea compared to when it is viewed eccentrically. As a result, the highest cFFFs are found when light hits the fovea directly [70]. Therefore, it is important to maintain both the distance between the light source and the observer’s eyes as well as the visual angle throughout the experimental protocol, if possible.

In open-water studies, results may be affected by differences in the positions of the light source and the observer’s eyes. For example, during scuba diving, measurements are taken on the surface as well as at varying depths. Additionally, movement of the light source or the observer can also increase cFFFs.

Eye motion due to saccades can also introduce errors in cFFF measurements. Humans are capable of perceiving flicker frequencies as high as 2 kHz during saccades [71].

Light source: The relationship between flicker frequency and flicker fusion was first described in a thesis [72]. The Ferry-Porter law later established that cFFF increases with luminance [73]. It was later found that cFFF also depends on the surrounding illumination [68]. This means that substantially higher rates of flicker can be seen under bright daytime illumination compared to dusk [74]. The adaptation of the eye, specifically the pupillary diameter, is also important [75,76]. For instance, cFFFs were about 5 Hz lower in dark-adapted than in light-adapted subjects [77].

However, few studies specify the illumination conditions, such as a study in which volunteers were dark-adapted for 15 min [22]. Eye adaptations could be a source of variation in cFFFs in literature, particularly when comparing measurements taken above sea level versus those taken at the seabed. Brighter stimuli have higher cFFFs, for example, and rod cells have less ability to achieve fusion than cone cells.

Aside from intensity, the wavelength (colour) of the light also influences cFFF [68]. The cFFF for red is significantly lower than for other colours [78], and the threshold for blue is significantly lower than for green [27]. Thus, comparing different studies becomes challenging when different colours are used. Moreover, the size of the light source also affects cFFF thresholds [22,79]. Lastly, the light-dark ratio of the stimulus also makes a significant difference [80,81].

We are confident that the three confounding factors mentioned above, namely colour, stimulus area, and light-dark ratio, have contributed to different cFFF thresholds observed under control conditions, which averaged 44 Hz [42], 35 Hz [43], and 30 Hz [82].

Frequency Change Direction: In many studies on cFFF, measurements usually begin at low flicker frequencies and gradually increase thereafter. The technique used to change frequencies, whether manual or automatic, does not seem to significantly affect results, as both techniques differ by a maximum of 2.2% [42]. However, there are few studies where frequencies were both increased and then decreased. In one group of “unsure” volunteers, there were no differences observed between increasing or decreasing flicker frequencies. In contrast, a group of “reckless” volunteers showed significantly higher descending frequencies than when frequencies were increased. This contradictive result could be interpreted based on possible personality-dependent differences [83].

Minor differences were found in a group of bicyclists depending on the direction of changing the flicker frequency [84], and the maximum differences reported in a group of yoga participants were about 4% [85].

It is not surprising that the order of changing the frequency matters, as it is consistent with perceptual constancy [86]. This means that an impression tends to conform to the object as it is or is assumed to be, rather than to the actual stimulus, such as flickering light or steady light.

A suggestion is to avoid using the term “fusion” when the frequency is decreased within a protocol and replace it with “separation”, for example, because “fusion” implies constancy.

Another aspect worth mentioning is the learning effect observed in cFFF measurements. The cFFF of the second test run was found to have increased significantly over the first run by roughly 3% [55]. Therefore, if the first test run was performed during control and the second during or after an intervention, a difference might have been introduced by the learning effect.

### 2.4. Environment

Stress can affect the cFFF threshold in various ways. For instance, psychological stress and anxiety can cause a significant decrease in cFFF both before and after surgery [87]. A similar decrease in cFFF was observed in individuals with insufficient sleep compared to those with sufficient sleep patterns and duration [20,22].

Yoga has been shown to increase cFFF values by 15% in participants who underwent a 30-day course. This increase is believed to be due to the reduction of physiological signs of stress, such as heart rate, breathing rate, and oxygen consumption [85]. Conversely, pharmacological means of reducing stress through sedatives can decrease both cFFF and cognitive competence [88,89].

Diving can also increase stress, especially for beginners or in adverse circumstances [46,90]. In such cases, cFFF values may decrease, and it can become challenging to distinguish between the effects of stress and those of toxic breathing gases.

Fatigue: It is worth noting that fatigue can also influence cFFF values. While cFFF is not considered a sensitive indicator of general fatigue [59], it can be used to measure central fatigue [10,11]. In one study, cFFF values were significantly lower in railway dispatchers after night shifts, suggesting that cFFF can also measure the degree of fatigue in the central nervous system [91]. However, after a maximum incremental cycle ergometer test on professional cyclists, cFFF thresholds increased, while the cyclists reported marked subjective fatigue, which was likely more peripheral than due to central nervous system fatigue [84]. Finally, even low doses of sedatives can lead to cFFF decreases and significant changes in subjective reports of fatigue [89].

Temperature: After a 90-min exposure to ambient heat, cFFFs were found to be increased in young army men [92]. Furthermore, a 0.5 °C increase in sublingual temperature was accompanied by a 6% improvement in flicker threshold [93]. Similarly, in young volunteers, an increase in core body temperature by 1.1 °C through the use of cling film, tin foil, and warming blankets resulted in increased cFFFs. The authors concluded that increased core body temperature was associated with improved temporal visual resolution and retinal trunk vessel dilation. These findings suggest that hyperthermia is associated with enhanced retinal function and increased retinal metabolism [94].

Temperature changes in either direction affect cFFF thresholds. Considering the aforementioned results, it is worth investigating hyperbaric chambers, where pressure increases are often associated with temperature increases. During a 60-min dive to 46 m using trimix 19/40, the skin temperature of participants decreased from 33 to 28 °C due to water temperature of only 4 °C. This decrease in skin temperature led to significantly increased cFFF readings [14]. Not only were the underwater cFFF measurements conducted in a different environment than the two surface measurements, but different optical media were also employed, the possible effects of which might have acted as a confounder and thus, contributed to an increased cFFF threshold.

Another study provides a possible explanation. Volunteers immersed their forearms in water of different temperatures, and the cFFF threshold was higher in cold water than in lukewarm water, indicating increased alertness from the cold stimulus. However, short-term memory was attenuated in the cold-water condition [95].

Time. The cFFF exhibits diurnal variation by decreasing throughout the day, which has been interpreted as a reduction in perceptual sensitivity [96]. Fortunately, subjects who undergo cFFF testing at the same time on sequential days show stable thresholds. However, if new activities are started, such as 1 h of motion training per day for nine days, participants show a 30% increase in cFFF [97].

Thus, any effect of daytime or lifestyle should be excluded, as the perceptual experience of subjects can dramatically alter the cFFF thresholds and should be a vital consideration in the control of studies employing the cFFF as a measure [97].

Another time aspect that might distort results is related to sequential cFFF measurements in a group of participants. For instance, in a multiplace dry chamber with twelve participants subjected to the same protocol, the investigator assesses the cFFF of the first participant and then continues with the other eleven. As a result, the last participant will be exposed to the pressure of 6 bar for a considerably longer time than the first. If each measurement should last 30 s, then the maximum delay will amount to 6 min., Decompression of the first participant will therefore be much shorter at the time of the cFFF measurement. A comparable situation exists with in-water dives. In a group of three divers, one might act as the investigator, another as an observer, and the third will write down the result. Thereafter, the tasks rotate until each diver has taken on one of the three tasks. It is unlikely that each diver will complete their task in precisely the same amount of time, and those who take longer would naturally be exposed for a longer period of time. Hence, the individual cFFFs will depend on the individual time of exposure.

### 2.5. Breathing Gases

In line with the lipid theory [98], it has been shown that inert gas narcosis (IGN) can impair cognitive performance [49,99]. As a result, it is expected that cFFFs will decrease at depths when partial nitrogen pressures are increased. Surprisingly, a study by [42] found that cFFF did not significantly change when breathing air at 6 bar in a dry chamber. However, a different study suggests that IGN can have a lasting effect on cognitive impairment, as assessed by cFFF, for at least 30 min after surfacing [44]. This latter study also suggests that IGN may depend on a gas-protein interaction, which is different from the lipid theory. According to the protein theory, inert gases act by binding to neurotransmitter protein receptors (for review, see [100]).

Similarly, studies on the effects of oxygen (O_2_) have yielded inconsistent results. One study found that breathing O_2_ at 1 bar and 2.8 bar caused a significant decrease in cFFF [42]. On the other hand, the effects of O_2_ on neuronal excitability in young healthy men were described as dose-dependent: 0.7 bar O_2_ did not affect cFFF, 1.4 bar O_2_ significantly jeopardized cFFFs, while 2.8 bar O_2_ allowed for recovery of cFFF [50]. These results on oxygen are concerning, especially considering that normobaric oxygen is widely used in clinical settings [101,102], and a partial pressure of 1.4 bar is considered a safe upper limit for divers [103].

In one additional study conducted at 6 bar in a hyperbaric chamber, two reaction time tests showed a deterioration in cognitive competence, while the cFFF was unexpectedly increased. The authors suggest that divers susceptible to IGN may also be susceptible to the effects of elevated partial pressure of O_2_, which could explain these counterintuitive results [104].

## 3. Conclusions

The first use of a cFFF test in the field of diving dates back almost 50 years [41], and it was rediscovered only a decade ago [44]. Since then, the test has yielded several valuable but also inconsistent results, raising questions about how to interpret some of them.

For example, a 15% increase in cFFF was observed after a yoga course [85], while participants in a daily 1-h motion training for nine days showed a cFFF increase of 30% [97]. These positive effects are easily understandable. However, a study on O_2_ effects found significant cFFF differences between control (1 bar O_2_) and intervention (2.8 bar O_2_) of only <3% [42], leaving interpretation open to question. Our pilot study with up- and down-flicker frequency changes provided significant cFFF differences of 6% at a *p*-value < 0.05 [105]. However, statistical significance based on a *p*-value does not measure the effect size or importance of a result [106]. To better understand our result, we calculated Cohen’s d [107], which showed that our result signifies only a “low effect”, despite a significant difference of 6%.

To emphasize our caution with *p* < 0.05 results, we refer to data from another study, where significant cFFF differences between 1 bar and 6 bar in a hyperbaric chamber averaged 0.9 Hz [14]. Calculating Cohen’s d again yielded a value of 0.46, signifying a “low effect”.

We believe that accurate interpretation of cFFF thresholds will remain an important diagnostic tool for eye diseases, as well as neurological and internal diseases. Furthermore, understanding intervention-induced cFFF changes can enhance our understanding of their significance. Despite the presence of several confounding factors, the use of cFFF techniques to evaluate cerebral impairment in humans, particularly in divers exposed to different breathing gases and dive profiles, is highly valuable, especially because divers may not subjectively experience narcosis.

Erlick and Landis’ (1952) [52] early observation of more than ten factors that might confound “true” cFFF values has been borne out. The cFFF technique is generally considered a simple, rapid, reliable, and reproducible tool in the field of experimental research but its validity will be compromised in the presence of confounding factors. The interpretation of results with the help of *p*-values seems questionable when significant differences are in the range of a few percent. It is important to note that confounding factors can lead to significant differences within open-field studies. Carefully considering the numerous confounding factors when utilizing cFFF thresholds to assess cerebral impairment is crucial in enhancing diving guidelines and safety.

## Figures and Tables

**Figure 1 medicina-59-00800-f001:**
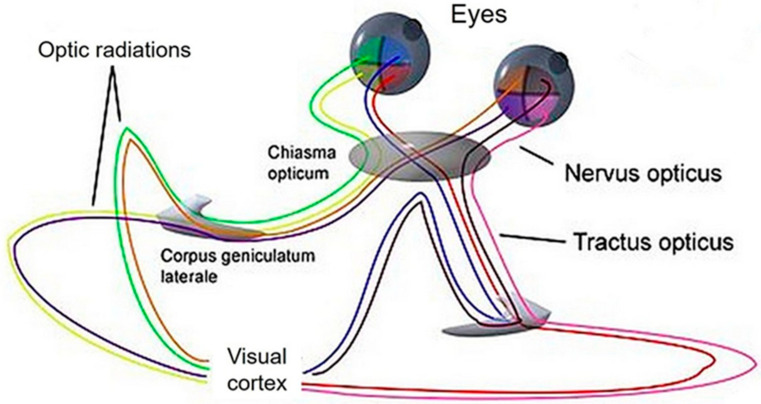
Visual pathway. The visual pathway starts when light passes through the cornea and lens to reach the retina, and continues through the optic nerve, optic chiasm, optic tract, lateral geniculate body and optic radiation until it reaches the primary visual cortex. https://de.wikipedia.org/wiki/Sehbahn (accessed on 5 April 2023).

## Data Availability

In this narrative review, no new data were created.

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
