# Peer review of "Assessing Critical Flicker Fusion Frequency: Which Confounders? A Narrative Review"

_medicina, 2023, doi:10.3390/medicina59040800_

Round 1
Reviewer 1 Report
The authors carried out a narrative review on a tool used in numerous studies and which indeed requires great vigilance in its interpretation. An inventory of all the parameters influencing this cFFF is therefore not useless. The review is well referenced and touches on a large number of areas in which cFFF is used. However, I have a few comments:
- As the review is addressed to a medical journal, ophthalmology section, the take-away message should be more general and less centered on diving. It is obvious that many studies in diving or hyperbaric chamber have been carried out with the cFFF which is an interesting tool, but all the personal additions of the authors are also directed towards diving. However, in some places, it suffices to change "a diver" to "an athlete" or "a subject".
If, on the other hand, it is the intention of the authors to insist on the fact that the cFFF would be a tool making it possible to contribute to the improvement of the guidelines in diving as explained in the conclusion, it would be good to point this out in the objective of the narrative review.
- Since the variations in absolute cFFF results are so great between subjects (due to gender, personality, age...), I am surprised that throughout the review and certainly in the conclusion, the authors have never spoken of the importance of taking a subject as its own control in order to already eliminate all the inter-personal variations. If the aim of the review is to highlight all the parameters that influence cFFF, it seems to me that the conclusion must insist on the precautions to be taken in the implementation of research protocols and in the interpretation of results. Again, for my part, a take-away message is missing.
Author Response
Reviewer #1
The authors carried out a narrative review on a tool used in numerous studies and which indeed requires great vigilance in its interpretation. An inventory of all the parameters influencing this cFFF is therefore not useless. The review is well referenced and touches on a large number of areas in which cFFF is used.
Thank you for your positive and constructive evaluation.
However, I have a few comments:
- As the review is addressed to a medical journal, ophthalmology section, the take-away message should be more general and less centered on diving.
The take-away message – our conclusions – are more general, by now.
It is obvious that many studies in diving or hyperbaric chamber have been carried out with the cFFF which is an interesting tool, but all the personal additions of the authors are also directed towards diving. However, in some places, it suffices to change "a diver" to "an athlete" or "a subject".
We followed your suggestion and have replaced diver where appropriate.
If, on the other hand, it is the intention of the authors to insist on the fact that the cFFF would be a tool making it possible to contribute to the improvement of the guidelines in diving as explained in the conclusion, it would be good to point this out in the objective of the narrative review.
Thank you for this comment. The last para of the Introduction section does consider that comment.
- Since the variations in absolute cFFF results are so great between subjects (due to gender, personality, age...), I am surprised that throughout the review and certainly in the conclusion, the authors have never spoken of the importance of taking a subject as its own control in order to already eliminate all the inter-personal variations.
In the initial draft, a section had been left out that considered the inter-personal variations. This data was available but had not been included.
If the aim of the review is to highlight all the parameters that influence cFFF, it seems to me that the conclusion must insist on the precautions to be taken in the implementation of research protocols and in the interpretation of results. Again, for my part, a take-away message is missing.
The Conclusion section is now considerable extended, thus considering your comment.

Reviewer 2 Report
The manuscript titled "Assessing critical flicker frequency: which confounders? A narrative review" by Muth et al., is an interesting review. However, it requires to be revised by the Authors as follows:
1- The English style and syntax need an extensive revision to make this manuscript clearer to readers.
2- The "discussion section" should be better organized and named differently. The word "discussion" doesn't seem appropriate.
3- The Authors can better discuss the impact of the cFFF test related to eye disease diagnosis.
4- The title should be corrected as the word “fusion” is missing.
Author Response
Reviewer #2
The manuscript titled "Assessing critical flicker frequency: which confounders? A narrative review" by Muth et al., is an interesting review. However, it requires to be revised by the Authors as follows:
Thank you for this general evaluation. In the following, we will address the four aspects that you mention.
1- The English style and syntax need an extensive revision to make this manuscript clearer to readers.
We have thoroughly revised the manuscript with regard to the usage of the English language. We believe that these changes have significantly improved the clarity and readability of the article. As a result, we are confident that our work is now much clearer and more accessible to readers. We appreciate your feedback, which helped us to identify areas for improvement
2- The "discussion section" should be better organized and named differently. The word "discussion" doesn't seem appropriate.
We have renamed the former discussion section to "Confounding factors," and introduced section titles and subtitles to better organize the content.
3- The Authors can better discuss the impact of the cFFF test related to eye disease diagnosis.
Thank you for your interesting suggestion. We strongly believe that the topic you proposed, cFFF and eye disease, deserves its own review article. To ensure that readers are informed about our intentions from the outset, we have included a mention of the topic diving and hyperbaric medicine in both the abstract and introduction sections of the manuscript.
4- The title should be corrected as the word “fusion” is missing.
Done.
Reviewer 3 Report
Dear authors,
In this study the authors assess critical flicker frequency. Although the study has the potentiality of being shared with the scientific community, I believe that the manuscript would benefit from a minor revision with the attempt to better support their experimental setting.
1. The theoretical framework is scarce, they should clearly describe the scientific evidence that supports the hypothesis they have raised.
2. The Discussion should be enriched with the existing theory. The authors should clearly describe the scientific evidence that supports their findings. In addition, they should start with a first paragraph describing the main aims and then the main results.
Kind regards
Author Response
Reviewer #3
Dear authors,
In this study the authors assess critical flicker frequency. Although the study has the potentiality of being shared with the scientific community, I believe that the manuscript would benefit from a minor revision with the attempt to better support their experimental setting.
Dear reviewer,
Thank you for your valuable feedback on our review article. We extensively searched for articles related to critical flicker fusion frequency (cFFF) and factors that could potentially confound the respective thresholds. The experimental settings and individual clinical, laboratory and in-field experimental details can be found in the published articles we referenced. However, we apologize for not providing a description of the experimental settings used by the other authors as well as our own.
- The theoretical framework is scarce, ….
In order to facilitate the understanding of our review article, we have included a figure describing the visual pathway involved in the perception of flicker light. This figure illustrates the anatomical structures that play a role in the processing of flicker light, and we hope it will help to clarify our discussion of critical flicker fusion frequency and related factors.
Moreover, we have added a new paragraph discussing one theoretical aspect of assessing critical flicker fusion frequency. Specifically, we highlight the theory of signal detection (Green & Swets, 1966), which provides a framework for differentiating between a participant's ability to distinguish between events and motivational effects or response biases (Pastore & Scheirer, 1974). We believe that this theoretical perspective is an important aspect of understanding the factors that influence flicker perception and will be useful for readers interested in this topic.
.….. they should clearly describe the scientific evidence that supports the hypothesis they have raised.
Throughout our review article, we highlight the value of the cFFF technique for assessing various aspects of flicker perception. However, we also acknowledge, i.e. we have the hypothesis, that the thresholds obtained through this technique can be confounded by a variety of factors. In fact, we identify and discuss over ten such confounders in the latter part of our review.
- The Discussion should be enriched with the existing theory. The authors should clearly describe the scientific evidence that supports their findings. In addition, they should start with a first paragraph describing the main aims and then the main results.
For this review article, we have searched the literature related to the cFFF and included over a hundred articles on that topic. In the Introduction section, we now explain our major aim, namely description of the various cFFF confounders. After categorizing them, we present such confounders in the Main section and draw conclusions towards the end of this review article.
Round 2
Reviewer 2 Report
The manuscript titled "Assessing critical flicker frequency: which confounders? A narrative review" by Muth et al., is an interesting review and the Authors have addressed the main points I have indicated in the previous report. The manuscript can be accepted for publication after minor revisions as follows:
1) The title should be correct as the word "fusion" is missing".
2) Lines 54-55: Please re-write this sentence to make it clearer to the reader.
3) Line 108: Please define abbreviations upon first appearance in the text.
4) Line 134: Please change "The" study to "This" study.
5) Line 136: Please change "the" review to "this" review.
6) line 154: Please change " an own" to "our".
7) The postscript section should be combined with the conclusion as a "Concluding Remarks" section.
8) Line 54: Please change "pateints" to "patients". Spell check required for the entire manuscript.
Author Response
The manuscript titled "Assessing critical flicker frequency: which confounders? A narrative review" by Muth et al., is an interesting review and the Authors have addressed the main points I have indicated in the previous report. The manuscript can be accepted for publication after minor revisions as follows:
Thank you for thoroughly reading the ms and your suggestions that were all addressed. Changes in the ‘red’ portions of the last version of the ms are highlighted in yellow.
- The title should be correct as the word "fusion" is missing".
done
- Lines 54-55: Please re-write this sentence to make it clearer to the reader.
Has been rephrased
- Line 108: Please define abbreviations upon first appearance in the text.
Scuba and CCR are defined.
- Line 134: Please change "The" study to "This" study.
done
- Line 136: Please change "the" review to "this" review.
done
- line 154: Please change " an own" to "our".
done
- The postscript section should be combined with the conclusion as a "Concluding Remarks" section.
done
8) Line 54: Please change "pateints" to "patients". Spell check required for the entire manuscript.
done